# Thermal Infrared Radiation and Laser Ultrasound for Deformation and Water Saturation Effects Testing in Limestone

**Alexander Kravcov [1,](ID), Elena Cherepetskaya [2], Pavel Svoboda [1], Dmitry Blokhin [2], Pavel Ivanov [2](ID) and Ivan Shibaev [2]**

[1] Department of Construction Technology, Faculty of Civil Engineering, Czech Technical University in Prague, 166 29 Prague, Czech Republic; pavel.svoboda@fsv.cvut.cz
[2] National Mining Institute, University of Science and Technology MISiS, 119049 Moscow, Russia; eb.cherepetskaya@misis.ru (E.C.); dblokhin@yandex.ru (D.B.); pn.ivanov@misis.ru (P.I.); shibaev@misis.ru (I.S.)
* Correspondence: kravtale@fsv.cvut.cz; Tel.: +420-224-355-461

**Abstract:** During the operation of engineering structures made of natural stone, for industrial and civil purposes, an important parameter in monitoring their technical condition is the assessment of their reliability and safety under the influence of various external influences. In this case, high-quality monitoring of the stress–strain state of natural stone structures, its physical, mechanical and filtration properties, as well as internal structural features is necessary to study the possibility of replacing individual elements of objects that have lost their original characteristics. To assess the state of geomaterials, this article proposes using a complex of introscopic methods, including infrared radiometry and laser-ultrasound structuroscopy. An important aspect is the calculation based on the Green–Christoffel equation of the velocity of a quasi-longitudinal wave in limestone consisting of densely packed, chaotically oriented calcite grains with a small quartz content. For the first time, using laser-ultrasonic structuroscopy and standard methods for determining open porosity, both total and closed porosity were determined. This allowed us to find the values of specific heat capacities of dry and water-saturated samples. The obtained values are used to find the ratio of changes in the temperature of dry and water-saturated samples at the same stress values. The results obtained demonstrate the need to take into account changes in the intensity of thermal radiation of limestone with different moisture content under conditions of uniaxial compression, when identifying changes in the stress state of elements of stone structures in real conditions.

**Keywords:** infrared radiometry; laser-ultrasonic structuroscopy; limestone; stress–strain state; water saturation

## 1. Introduction

Throughout human history, natural stone has been widely used as a material for monumental architecture and sculpture [1–4] as well as for industrial and civil construction [5,6]. Examples are the Moscow Kremlin, Athena's Temple and Apollo's Temple in Syracuse, and Spirito Santo Church in Melilli, all made of limestone, and modern buildings and subway stations with walls made of marble, granite, sandstone, etc. Since historic and architecturally significant buildings require restoration and rehabilitation, it is particularly important to select appropriate natural stone samples from the deposits that are currently being developed. Not only should the decorative properties of stone be taken into account, but also the physical and mechanical parameters and structural features. Note that

limestone is most often used for the above purposes, extracted from quarries and mines all over the world. For example, there are 71 limestone quarries in France alone.

Engineering structures made of natural stone are exposed to weathering agents and dynamic loads caused by both natural and human factors (earthquakes, vibrations, etc.) [7–10]. Clearly, these factors negatively affect the stability of the structures, stimulating destructive mechanical processes in their material. Therefore, it is necessary to constantly monitor the porosity, water absorption, changes in elasticity moduli and other parameters of natural stone, especially with respect to historic and architecturally significant buildings.

Today, there are a wide variety of methods for studying the internal structure and stress–strain behavior of natural stone and structures made of it. These methods comprise destructive methods involving load testing under different loading conditions [11,12] and semi-destructive mechanical tests with simultaneous measurement of acoustic emission [13,14]. Non-destructive in situ and laboratory methods for inspecting natural materials are addressed in [15–32], including thermal control [16–20], multispectral optical remote sensing [21], ground penetrating radar [22,23], ultrasonic inspection [25,26], gamma-ray logging [27], terahertz spectroscopy [28], X-ray tomography [29,30], neutron radiography [31], and others [32].

At present, the most common methods are thermography [16–20] and different versions of ultrasonic inspection [25,26].

IR thermography, or IR radiometry, involves non-contact measurement of changes in the intensity of infrared radiation emitted by the surface of geomaterial. Two methods of IR thermography [20] are used to study the properties of rocks: active and passive ones. Active thermal control involves heating the sample by a heat source located on its front side. The thermal fields inside geomaterial are redistributed due to hidden defects. Recorded temperature anomalies are used to evaluate the structure and the porosity in igneous, metamorphic and sedimentary [33,34]. In [35], it is shown that this method allows the permeability of rocks to be evaluated as well. In [36], active pulsed infrared thermography is used to identify and qualitatively evaluate the salt content in the natural stone of historic buildings.

Passive thermal control mostly involves analyzing heat flows produced as a result of deformation of rocks [37–42]. In that case, the interpretation of thermal IR radiation measurements is based on the well-known thermodynamic effects: changes in the temperature of solid bodies during their adiabatic deformation ('thermoelastic' and 'thermoplastic' effects) and temperature dependence of the intensity of infrared radiation emitted by the surface of solids.

Thus, it is shown in [37–42] that IR radiometry is an efficient method to identify stages of deformation of geomaterials of different types and water saturation effects [43]. It is found that the intensity of radiation emitted by quartz syenite, fine-grained diorite, and quartz monzonite changes with increasing load: from 8.3 to 10.1 μm, 10.3 to 12.2 μm, and 13.0 to 15.1 μm, respectively [32]. It is also shown that at a relatively low loading rate, the temperature remains constant due to heat exchange with the environment.

In [38], it is experimentally found that as mechanical load increases, the intensity of IR radiation is redistributed between the spectral components in the wavelength region from 7 to 11 μm. Thus, authors [40,42] performed a quantitative analysis of the relationship between stress applied to quartz sandstone and IR radiation; they showed that the highest intensity of radiation per unit stress was observed in the wavelength range from 8.0 to 11.5 μm. In [41,42,44], it is found that the mineral composition of geomaterial significantly influences the frequency range, within which the most intense radiation is observed under loading conditions. It is shown in [41,42] that porphyrite granite with high feldspar content has a load-sensitive wavelength range from 8.4 to 10.6 μm and granite with high plagioclase content has a load-sensitive wavelength range from 8.2 to 11.7 μm [44]. The load-sensitive frequency band is related to the range of IR emission spectra of individual minerals.

Note that the above-described findings emerged from remote IR sensing, when the distance to the test sample was several tens of centimeters (for example, in [42] this distance was 80 cm). In that

case, it was necessary to perform complex calibration of the equipment before every measurement so that atmospheric effects could be taken into account. Due to the narrow frequency ranges used in the above-mentioned studies, it was impossible to fully take into account the vibrational and rotational levels of all minerals, gases, and liquids in pores. Nevertheless, this method is quite effective for locating possible defects and assessing the water content and stress-strain behavior of materials.

However, it would be more efficient to use this method together with ultrasonic diagnostics so as to comprehensively assess the condition and internal structure of geomaterials. Conventional ultrasonic flaw detectors and tomographs operate, as a rule, at a certain resonance frequency [24–26], which makes it difficult to determine the geometry and location of different-scale defects. The use of piezoelectric transducers exciting and receiving broadband ultrasonic signals results in a sharp decrease in radiated power and a significant decrease in sensitivity, which means that the dynamic range becomes narrower. In this respect, laser ultrasonic structuroscopy and tomography [45–47] seem promising for characterizing the internal structure, porosity, and local elastic properties of natural stone. As is shown in [45,46], the main advantage of these methods is as follows: generated powerful ultrasonic pulses have strictly controlled shape and both transmitted signals and signals reflected from heterogeneities are recorded by broadband piezoelectric detectors.

In this study, ultrasonic structuroscopy and IR-radiometry were used to examine the structure and properties of limestone and changes in these properties with changing uniaxial stress and water saturation.

## 2. Materials and Methods

### 2.1. Samples and Their Preparation

We examined samples of limestone, one of the most commonly used materials for construction. Limestone is a highly heterogeneous rock, so we chose samples with similar physical and mechanical parameters and structural features in order to ensure high-quality and reasonable results. For this purpose, a number of preliminary tests were carried out, including both standard ones for determining the physical and mechanical properties of rocks, and such introscopic methods as scanning electron microscopy and laser ultrasonic diagnostics.

First, we used a «Struers Labotom-15» cutting machine (Figure 1a) to saw one block of limestone into over a hundred rectangular parallelepipeds $25 \times 25 \times 50$ mm in size and then an automatic grinding polishing machine «Struers Tegramin-25» (Figure 1b) to make polished thin sections.

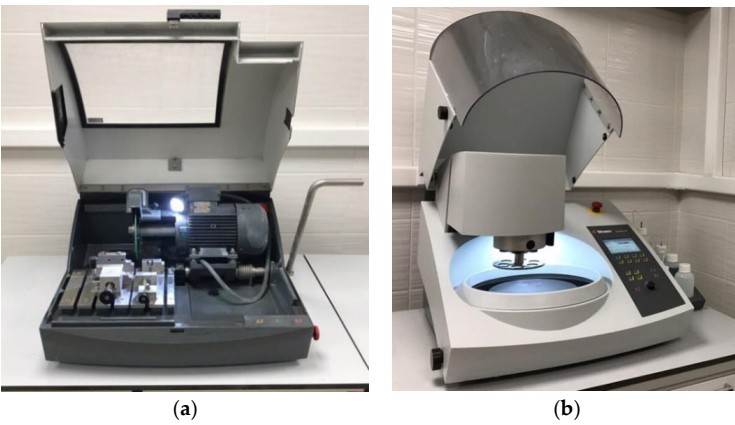

(**a**)　　　　　　　　　　　　　　　　　　(**b**)

**Figure 1.** «Struers Labotom-15» cut-off machine (**a**) and «Struers Tegramin-25» automatic grinding polishing machine (**b**).

Mineral and elemental analyzes were performed on this series of polished thin sections $25 \times 25$ mm in size using a «Phenom ProX» scanning electron microscope (Figure 2) operating in an optical imaging

mode for petrographic analysis and electronic imaging mode for chemical analysis based on energy dispersive system.

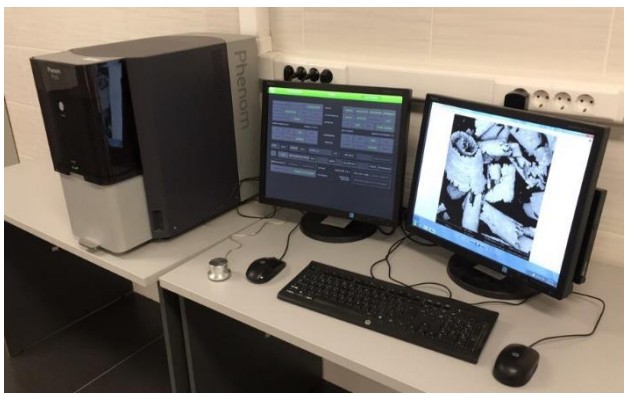

**Figure 2.** Scanning electron microscope «Phenom ProX».

It was found that most limestone samples contained 40.0–42.3% of calcium, 12.1–13.9% of carbon, and 45.1–47.5% of oxygen. Some insignificant mineral components and impurities, such as silicon (0.2–0.3%), magnesium (0.1–0.2%), and iron (0.1%), were also detected, distributed evenly on the surface of the samples.

The polished thin sections featured granular surfaces. Along grain boundaries, there were pore systems occupying 5–7% of the total surface area, their characteristic dimensions ranging from 20 to 40 µm. The mineral composition was represented by calcite (97.5–98.3%), quartz (1.5–2.3%), and dolomite (less than 0.5%). The total amount of other minerals was small (less than 0.1%).

Based on electron microscopy data, 25 samples out of the initial series were selected, having similar surface structural features. Similar surface structural features are common morphometric properties such as value of porosity and dimensions of pores.

### 2.2. Measurement of Local Elastic Wave Velocities in Samples by Laser Ultrasonic Structuroscopy

At the second stage, it was necessary to select samples with similar moduli of elasticity, without structural defects and with approximately the same porosity. This was done using laser ultrasonic structuroscopy [19]: 30 samples were examined with an UDL-2M automatic flaw detector. Figure 3 shows a schematic diagram of the detector.

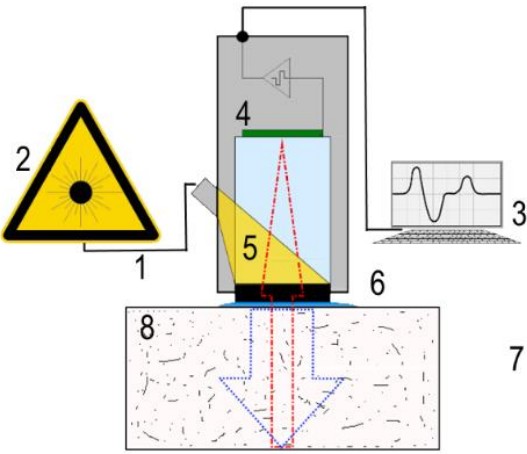

**Figure 3.** Schematic diagram of measurement of elastic wave velocities in limestone sample using UDL-2M laser ultrasonic flaw detector: optical cable (**1**), laser (**2**), computer (**3**), detector (**4**), laser radiation (**5**), optical-acoustic generator (**6**), pulses (**7**), rock sample (**8**).

The operating principle of the flaw detector UDL-2M is as follows: a special laser-based optoacoustic generator generates high-power broadband ultrasonic longitudinal pulses with a strictly controlled shape. The pressure amplitude distribution across the cross section of the acoustic beam is a Gaussian distribution; therefore, there is no noise interference in the form of side lobes in the radiation pattern. Consequently, the signal-to-noise ratio is higher, and the dynamic range is wider. Scattered, reflected, and transmitted signals are recorded with a broadband piezoelectric detector (a bandwidth of 100 kHz–20 MHz) combined with the generator. The aperture of the piezoelectric detectors is 4 mm. An ultrasonic signal 'cuts out' an elementary cylinder from the sample, with a diameter of 4 mm and length equal to the thickness of the sample. Longitudinal wave velocities Vi in every i-th cylinder (i = 1,2–100) are calculated and mapped, and the thickness of the sample and the double travel time of the acoustic pulse through the sample are taken into account.

Figure 4 shows two velocity maps derived from our experiments. Figure 4b shows a velocity distribution map for a sufficiently homogeneous limestone sample, velocities ranging from 4350 to 4650 m/s; the other sample—inhomogeneous (Figure 4a)—had defects and exhibited velocities varying from point to point between 4050 and 4600 m/s. Longitudinal wave velocities were determined in this mode with an error of 1%, therefore, those samples that had velocities changing from point to point by more than 5% were discarded [48].

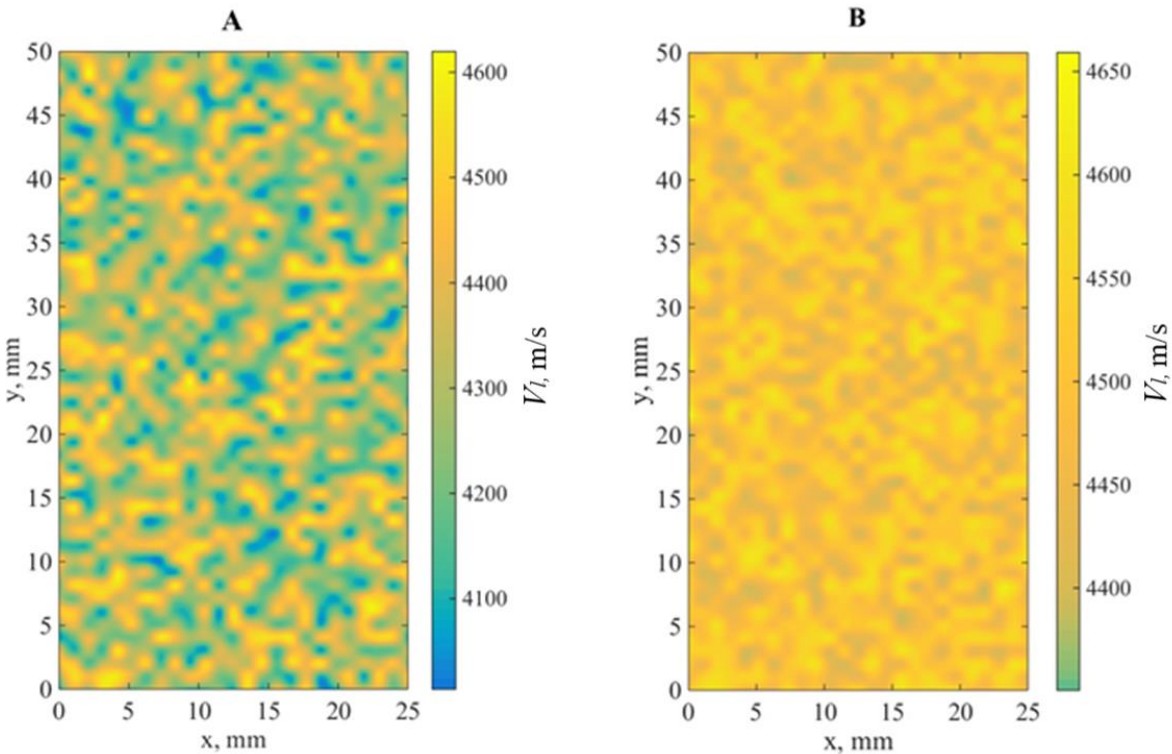

**Figure 4.** Longitudinal wave velocity distribution maps: inhomogeneous sample (**A**) and homogeneous sample (**B**).

Note that the reflection of longitudinal pulses from the opposite side of the sample produced shear wave whose time delay relative to the reference signal was used to determine its velocity $V_{ti}$ at every point. Figure 5 shows shear wave velocities. Shear waves were studied to ascertain that there were no cracks in the samples: they are more sensitive to the presence of defects than longitudinal waves [48].

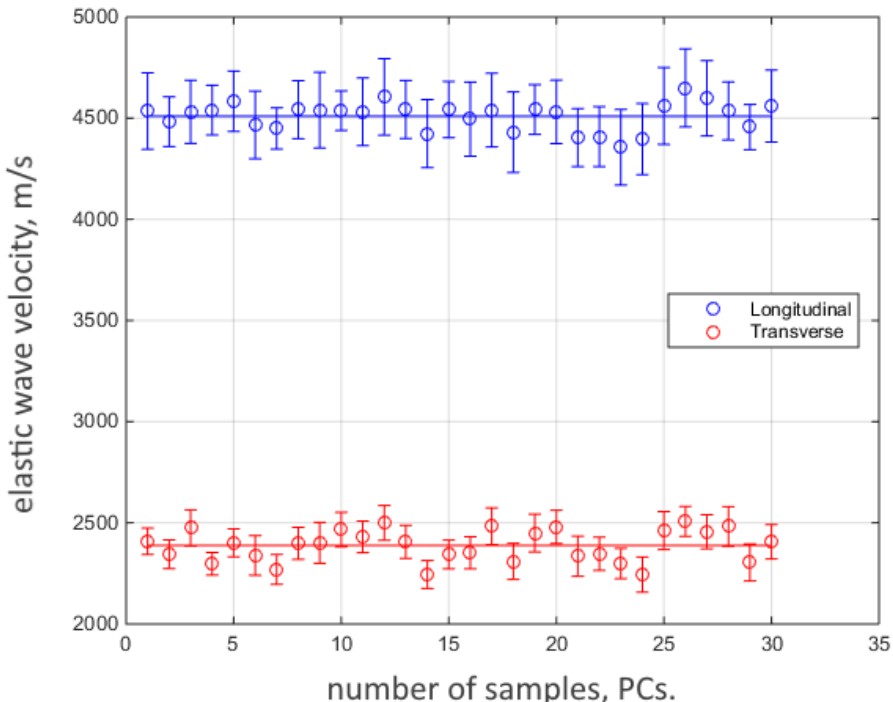

**Figure 5.** Average values of longitudinal and shear wave velocities in all selected samples.

*2.3. Calculation of the Total Porosity in Limestone Samples*

Based on the experimentally determined longitudinal wave velocities, we can calculate the average volumetric porosity P of every sample using the following expression [20]

$$P_{general} = \left(1 - \left(\frac{V_l}{V_0}\right)^2\right)^{\frac{3}{2}} \tag{1}$$

where $V_0$ is the longitudinal wave velocity in material with no pores ($Pi = 0$).

Porosity calculated by (1) is the total porosity (closed and open porosities). It was necessary to calculate $V_0$, for which the following algorithm was developed. It is well known that the principal minerals of limestone (calcite and quartz) belong to the trigonal symmetry class [47]. In trigonal crystals, purely longitudinal waves propagate only along three crystallographic axes |100|, |010|, and |001| their velocities, $V_1$, $V_2$, and $V_3$, are determined by the diagonal elements of the stiffness matrix $C_{11}$ and $C_{33}$ [49]

$$\rho V_1^2 = C_{33} \tag{2}$$

$$\rho V_2^2 = \rho V_3^2 = C_{11}, \tag{3}$$

where $\rho$ is the density of the crystal.

Quasi-longitudinal waves whose phase velocities are determined from the Green–Christoffel equation [49] propagate in all other directions; their velocities may differ significantly from those of pure modes. Since calcite and quartz are chaotically oriented in limestone, velocity $V_0$ should necessarily be calculated by averaging over all directions, which is a rather laborious procedure. Therefore, $V_0$ was estimated using the known coefficients of the stiffness matrix $\{C_{ij}\}, i, j = 1, \ldots, 6$ for calcite and quartz (see Table 1) to determine velocities $V_1$, $V_2$, and $V_3$, along the |100|, |010|, and |001| crystallographic axes and, additionally, velocities $V_{12}$, $V_{23}$, and $V_{13}$ in the |110|, |011|, and |101| directions for each mineral [50]. The calculation of velocity $V_{23}$ is given in Appendix A. Then, the velocities in all six directions were averaged, the contribution of calcite and quartz taken into account.

**Table 1.** Coefficients of the stiffness matrix for calcite and quartz.

| Mineral | $C_{11}$,GPa | $C_{12}$,GPa | $C_{44}$,GPa | $C_{33}$,GPa | $C_{13}$,GPa | $C_{66}$, GPa | $\rho$, kg/m³ |
|---------|---------|---------|---------|---------|---------|---------|---------|
| Calcite | 137 | 45.2 | 34.2 | 79.2 | 44.8 | 45.9 | 2980 |
| Quartz | 86.8 | 7.1 | 58.3 | 105.9 | −11.9 | 39.9 | 2650 |

It was found that the average velocity in limestone without pores was 4900 m/s; the average porosity in 30 selected samples was determined, taking expression into account (1). Figure 6 shows a histogram of the porosity distribution across the samples.

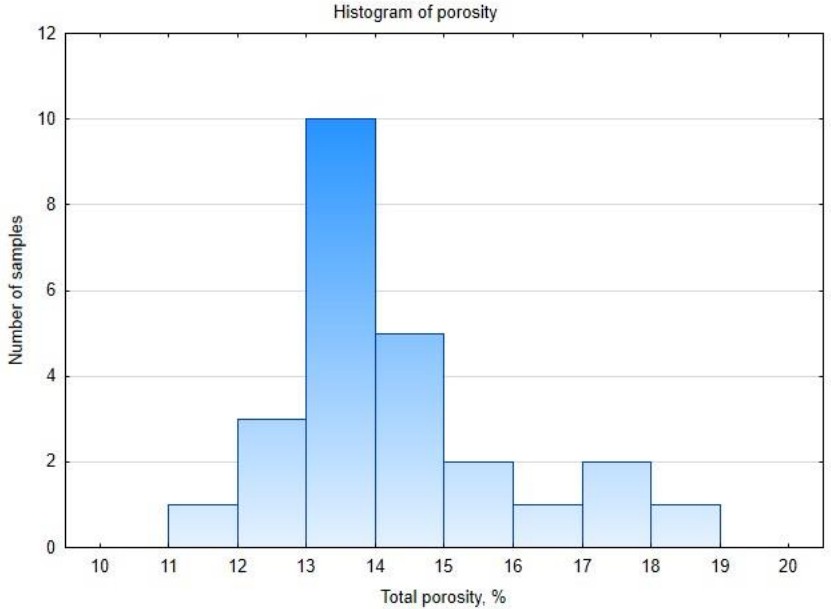

**Figure 6.** Porosity in limestone samples.

It is clear from Figure 6 that most limestone samples have porosity P ranging from 13 to 14%. That is why 10 samples with P = 13–14% were selected for mechanical tests. Mechanical tests were performed on five dry samples (Figure 7a) and five water-saturated samples (Figure 7b).

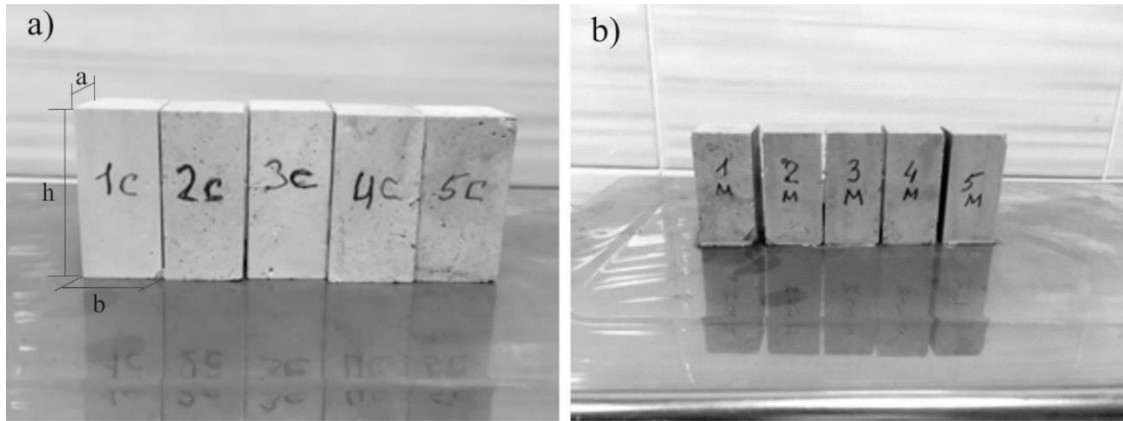

**Figure 7.** Limestone samples: dry samples (**a**) and water-saturated samples (**b**).

*2.4. Determination of Open Porosity*

To study the effect of water saturation on the elastic and thermal properties of the samples, it was necessary to estimate open porosity. Open porosity was determined by the Archimedes method

in accordance with ASTM C830-00 (2016) and BS EN 1936: 2006. First, the mass of dry samples was determined after drying in a vacuum oven at 105 °C for 24 h to completely remove residual moisture. After that, they were saturated with deionized water at atmospheric pressure (~0.1 MPa) for 48 h. Before weighting the water-saturated sample in air, excess water was removed from the surface with a damp cloth. The weight was measured using an electronic balance with an accuracy of 0.1 mg. The open pore volume $V_{open}$ was determined from the difference in weight between the dry and water-saturated samples, $m_{dry}$ and $m_{wet}$, respectively. Hereinafter, the index 'dry' refers to dry samples, and index 'wet' refers to water-saturated ones. The average weight of the dry samples was $m_{dry} = 67.14$ g and that of the water-saturated samples was $m_{wet} = 69.94$ g. The volume of open pores was calculated using the formula

$$V_{open} = \frac{m_{wet} - m_{dry}}{\rho_{wat}}, \tag{4}$$

where $\rho_{wat}$ is the density of water. Accordingly, the average open porosity defined as

$$P_{open} = \frac{V_{open}}{V_{sample}}, \tag{5}$$

was equal to 9.5%.

### 2.5. Mechanical Tests Accompanied by Infrared Radiation Measurements

The samples were subjected to uniaxial compression using an LFM-50 testing machine. The load rate of the samples was 0.28 kN/s, and longitudinal deformations were measured using the LDVT method of this machine. Deformation and IR radiation intensity were measured synchronously (Figure 8).

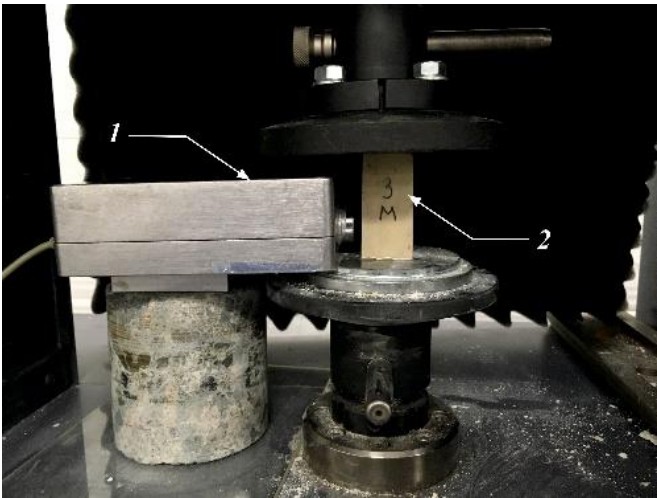

**Figure 8.** Experimental setup: IR radiation detector (**1**) and limestone sample (**2**).

An IR radiation detector based on a «RTN-31» detector [51] with a bandwidth from 3 to 14 μm was located facing the center of the sample at a distance of 0.5 cm from its surface. The wide frequency range made it possible to record the spectra of gases, liquids, and all minerals in the sample.

### 3. Results and Discussion

Figure 9 shows «$\sigma - \varepsilon$» plots derived from the above-described experiment. The $\sigma(\varepsilon)$ curves reflect the well-known fact that changes in the strength and deformation properties of limestone samples greatly depend on water saturation [25,27].

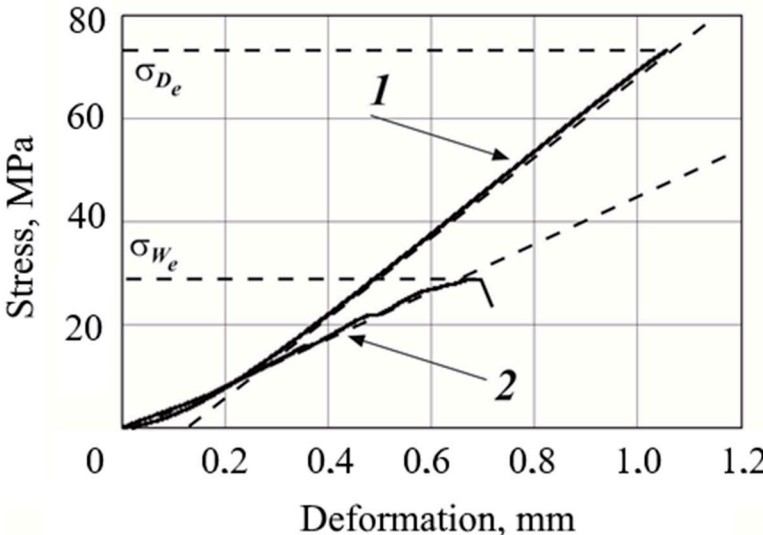

**Figure 9.** Plots «$\sigma - \varepsilon$» for dry (**1**) and water-saturated (**2**) limestone samples.

Figure 10 shows the dependence of the intensity of infrared radiation on deformation of dry and water-saturated limestone samples ($W_1(\varepsilon)$ and $W_2(\varepsilon)$, respectively) under uniaxial loading at a constant loading rate ($\frac{d\sigma}{dt} = const$). Note that the linear sections of the stress $\sigma$ vs. deformation $\varepsilon$ curves correspond to directly proportional dependences of $W_1(\varepsilon)$ and $W_2(\varepsilon)$ on $\varepsilon$.

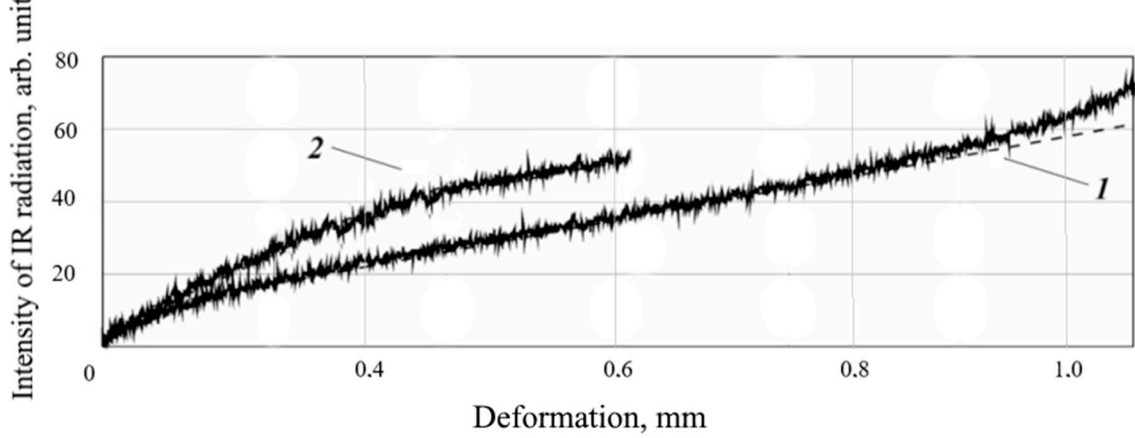

**Figure 10.** Intensity of infrared radiation vs. deformation: dry (**1**) and water-saturated (**2**) limestone samples.

Clearly, the inclination of the straight line approximating $W_2(\varepsilon)$ is significantly greater than that of the straight line approximating $W_1(\varepsilon)$, which indicates higher thermoactivity of water-saturated limestone. This is consistent with conclusions in [11,29]: the intensity of IR radiation from rock samples under compression increases with their water saturation. At the same time, it is mentioned that changes in the thermo-physical and physical and mechanical properties of samples under the influence of water saturation are the main factor causing the observed thermo-mechanical effect.

In order to verify the results, it is interesting to estimate temperature increments $\Delta T$ for dry or water-saturated samples. For this purpose, we apply the well-known approximation [52], which connects stress increments with changes in temperature during the uniaxial adiabatic straining of solid body

$$\Delta T = \frac{\alpha}{\rho \cdot c} T_0 \Delta \sigma, \tag{6}$$

where $T_0$ is the absolute value of temperature prior to deformation; $\alpha$ is the coefficient of linear expansion, $c$ is the specific heat at constant pressure; $\rho$ is the density of material.

Then, as follows from expression (6), the relationship between temperature changes in dry and water-saturated solid samples as a result of uniaxial adiabatic straining is as follows (provided that the samples exhibit the same stress increment $\Delta\sigma$)

$$\frac{\rho_{dry} \times c_{dry}}{\alpha_{dry}} \times \frac{\Delta T_{dry}}{T_0} = \frac{\rho_{wet} \times c_{wet}}{\alpha_{wet}} \times \frac{\Delta T_{wet}}{T_0}, \tag{7}$$

as is shown in [52], $\alpha^2 E = const$, where $E$ is the modulus of elasticity of material under uniaxial compression. Elastic moduli $E_{dry}$ and $E_{wet}$ of dry and water-saturated samples are calculated from the deformation curves (Figure 10): $E_{dry} = 69$ GPa and $E_{wet} = 42$ GPa. Taking into account these parameters, Formula (7) takes the following form

$$\Delta T_{wet} = \frac{\rho_{dry} \times c_{dry} \times \sqrt{E_{dry}}}{\rho_{wet} \times c_{wet} \times \sqrt{E_{wet}}} \cdot \Delta T_{dry}, \tag{8}$$

The densities and moduli of elasticity were measured. Therefore, in order to assess the relationship between temperature changes in dry and water-saturated limestone samples, it was necessary to calculate their specific heat capacities $c$. According to [9], the specific heat of a heterogeneous medium is the arithmetic weighted average of all mineral components with their share $k_i$ and specific heat $c_i$, that is

$$c = \sum c_i \times k_i, \tag{9}$$

Since the principal component of the limestone samples is calcite and there are pores, the following formula for the specific heat of water-saturated limestone is derived from (8)

$$c_{wet} = \sum c_i \times k_i = \frac{c_{cal} \times \rho_{cal} \times (1 - P) + c_{water} \times \rho_{water} \times P_{open}}{\rho_{wet}}, \tag{10}$$

The index "*cal*" refers to calcite grains. Similarly, in accordance with [9], the following expression is derived for the specific heat of dry limestone

$$c_{dry} = \frac{c_{cal} \times \rho_{cal} \times (1 - P)}{\rho_{dry}}, \tag{11}$$

In expression [11], the heat capacity of air in pores of dry limestone is not taken into account. Specific heat capacities of dry and water-saturated samples, calculated by Formulas (10) and (11), are presented in Table 2.

**Table 2.** Specific heat capacities of dry and water-saturated limestone samples.

| Dry Limestone Samples | | Water-Saturated Limestone Samples | |
|---|---|---|---|
| No. | $c$, J/(kg·K) | No. | $c$, J/(kg·K) |
| 1d | 1059 | 1w | 1149 |
| 2d | 1023 | 2w | 1177 |
| 3d | 1015 | 3w | 1131 |
| 4d | 1021 | 4w | 1182 |
| 5d | 1044 | 5w | 1185 |
| Average | 1032 | Average | 1165 |

The average values of specific heat capacities $\left(c_{dry}\right)$ and $\left(c_{wet}\right)$ of dry and water-saturated samples (see Table 2) are used in subsequent calculations by Formula (8), $T_0 = 300$ K at that.

Substituting the above values into (7), we find that the temperature increment for water-saturated samples is as follows: $\Delta T_{wet} \approx 1,2 \times \Delta T_{dry}$, which confirms the experimental findings presented in Figure 5.

Thus, our experiments and numerical evaluation show that the water saturation of porous materials significantly affects their mechanical and thermo-physical characteristics, which is manifested, in particular, in a significant increase in the thermal activity initiated by deformation processes.

## 4. Conclusions

Our study demonstrates an integrated approach to laboratory research into thermo-mechanical processes in complexly structured heterogeneous materials (limestone samples) under loading conditions.

Conventional methods of scanning electron microscopy were employed to perform petrographic analysis of limestone samples and reveal limestone structural features. It is shown that laser ultrasonic structuroscopy can be efficiently used to quickly evaluate the porosity in rocks.

It is found that the intensity $W(t)$ of thermal radiation emitted by the surface of limestone samples under uniaxial loading depends on the water content. Importantly, variations in the intensity of thermal radiation and changes in mechanical parameters were measured synchronously. Analysis of these measurements of axial stresses and strains showed the expected significant deterioration of the mechanical properties (i.e., ultimate strength under uniaxial compression and modulus of elasticity) of water-saturated samples as compared to dry ones. Analysis of the IR radiometric measurements shows that the nature of $W(t)$ unambiguously depends on the water saturation of limestone samples, which means that the revealed regularity should be taken into account when monitoring and evaluating changes in the stress–strain behavior of stone elements of constructions under real-life conditions.

**Author Contributions:** E.C., D.B. and P.S. developed the conceptualization of this work; A.K. designed the experiments; P.I. and I.S. performed the experiments; all the authors participated in the data processing and the analysis of the results. All authors have read and agreed to the published version of the manuscript.

**Funding:** This research was funded by NUST MISIS Competitiveness Program, grant No. K2-2019-004.

**Acknowledgments:** Authors are grateful to the Czech Technical University in Prague and the NUDB, Czech Republic for its financial support. They also want to thanks to Vladimir Křistek from CTU in Prague for his coordination and kind advices.

**Conflicts of Interest:** The authors declare no conflict of interest.

## Appendix A

*Calculation of Quasi-Longitudinal Wave Velocities in Calcite*

The conditions of the existence of eigenvalues of phase velocities and eigenvectors with components $U_l = (l = 1, 2, 3)$ in quasi-longitudinal and quasi-transverse waves propagating in an arbitrary direction are determined using the Green–Christoffel equation

$$\left(\mathbf{\Gamma}_{il} - \rho \times V^2 \times \delta_{il}\right) \times U_l = 0 \tag{A1}$$

where $\rho$ is the density of material, $V$ is the phase velocity, $\mathbf{\Gamma}_{il}(i = 1, 2, \ldots, 6)$ is the Christoffel tensor, $\delta_{il}$ is the Kronecker symbol.

System (A1) has a unique nontrivial solution if the determinant composed of the coefficients at $U_l$ is equal to zero

$$\left|\mathbf{\Gamma}_{il} - \rho \times V^2 \times \delta_{il}\right| = 0 \tag{A2}$$

Equation (A2) in the general case is a cubic equation with respect to $\rho \times V^2$.

For the |110| direction of the trigonal symmetry class, the components of the Christoffel tensor are as follows

$$\Gamma_{11} = C_{11} + C_{66} \quad \Gamma_{12} = C_{12} + C_{66}\Gamma_{22} = C_{66} + C_{11} \quad \Gamma_{13} = 2C_{14}\Gamma_{33} = 2C_{44} \quad \Gamma_{23} = -C_{14}$$

Then, the determinant (A2) is as follows

$$\begin{vmatrix} C_{11} + C_{66} - \rho \times V^2 & C_{22} + C_{66} & 2C_{14} \\ C_{12} + C_{66} & C_{66} + C_{11} - \rho \times V^2 & -C_{14} \\ 2C_{14} & -C_{14} & 2C_{44} - \rho \times V^2 \end{vmatrix} = 0 \tag{A3}$$

This determinant is equivalent to an cubic equation with respect to $\rho \times V^2$, the roots of which specify the phase velocities of two quasi-transverse waves and one quasi-longitudinal wave, to which the greatest value corresponds. This equation was numerically solved; it was found that the quasi-longitudinal wave velocity was 4900 m/s in the |110| direction.

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
