# Peer review of "Thermal Infrared Radiation and Laser Ultrasound for Deformation and Water Saturation Effects Testing in Limestone"

_remotesensing, doi:10.3390/rs12244036_

Round 1

Reviewer 1 Report

  • Correction is required after checking the numbering of subtitles 2.3 and 2.4.
  • In the 82th sentence, it is necessary to change from word “autors” to "authors".
  • In Fig. 5, the unit description for the X and Y axis of the graph is required.
  • When measuring the radiation intensity, did not the result data have much influence due to reflection or scattering caused by water-saturated?
  • In the 281th sentence, it is necessary to check the decimal point for the relational expression.
  • In the sentence 279th, it is necessary to check “Table 5”
  • English language editing

   There are some grammatical mistakes and drawbacks in the manuscript, please check

the Grammarly and try to present a concise expression.

Author Response

Thank you for your comments. The adjustments have been made. It is important to note that when editing an article, the numbering of the "lines" changed. Major fixes have been edited / added and are highlighted in green.

It is also important to note that the abstract was formulated with an emphasized wording of novelty and specific contributions from this article.

Brief comments on the comments of Reviewer 1.

  1. Adjustments to subheadings 2.3 and 2.4 have been made;
  2. The word "autors" to "authors" edited;
  3. In Figure 5, the axes are labeled, the units of measurement are marked;
  4. The water saturation of the sample influenced the radiation intensity, the data is shown in Figure 10;
  5. 281 suggestions checked and adjusted;
  6. The table has been checked, the table number has been changed to the correct one.

Reviewer 2 Report

This paper deals with the application of IR-radiometry and UT for the assessment of limestone under various conditions. This research is in a very interesting topic, but the methodology lacks novelty and this research does not add any knowledge to the field. The ultrasonic structuroscopy, as it is named by the authors, has already been presented in previous work and the IR radiometry is a very well documented method applied for the monitoring of materials similar to this research.  

Initially, the abstract needs to be thoroughly reviewed as it has some grammatical errors and does not provide clear information to the reader regarding this specific research. The introduction provides sufficient background and include relevant references.

The section 2 “Materials and Methods” needs improvement as there is lack of information regarding the methodologies.

Line 140: The authors should clarify and analyse the “similar surface structural features”.

Line 144: More information regarding the UDL-2M flaw detector should be added. Lines 157-158: The text contradicts with the Figure 4. Is 4b inhomogeneous or homogenous?

Line 161: The authors should explain why the error is 1%. At the same line they should also explain why velocity changes more than 5% error were discarded. Is this a standard practice? If yes, the authors should include evidence from literature.

Figure 4: Units should be added for the velocities.

Lines: 173-174 The authors are asked to support their statement: “they are more sensitive to the presence of defects than longitudinal waves” using literature resources.

Line 225: More information (eg. capacity, type, loading/strain rate etc.) regarding the compression test should be provided by the authors.

Line 229: According to reference 50, ir detector was RTN-31, is this the same detector with the PTH-31? More information should be included about the detector and the IR radiometry.

Line 234: What is σ(ε)? Do you mean the stress strain curves (?−ε)?  

Figure 9. Are these typical plots? Please use comma for decimals.

Comparing the results of Fig 9 and 10, there is a discrepancy for sample 1. In Figure 9 the samples failed at 0.7mm deformation but the IR data in Fig 10 are up to 0.61mm. Could the authors explain the reason of this?

Line 261: Under what conditions this expression is constant?

Equation 8. The authors should explain the derivation of eq.8.

Line 265: How the densities and E have been measured? The authors are asked to explain.

The manuscript has limited amount of results and they lack novelty.

The authors have used either conventional methods or methods that have been presented in previous published work. The authors claimed that this study demonstrates an integrated approach applied to complexly structured materials. Limited variety of complex structures/samples has been presented and analysed in this work and the integrated approach is not evident in the manuscript.

Author Response

Thank you for your comments. The adjustments have been made. It is important to note that when editing an article, the numbering of the "lines" changed. Major fixes have been edited / added and are highlighted in green.

It is also important to note that the abstract was formulated with an emphasized wording of novelty and specific contributions from this article.

Replies to the comments of Reviewer 2.

  1. Clarification of the term “similar surface structural features” added;
  2. Information about the flaw detector is indicated in the next paragraph after its mention;
  3. The uniformity of the samples has been clarified, the maps have been adjusted;
  4. The accuracy of determining the velocities using a laser ultrasonic flaw detector has been clarified with reference to article [52];
  5. In Figure 4, added units of speed measurement;
  6. "lines: 173-174" - Vasily Zarubin, Anton Bychkov, Vyacheslav Zhigarkov, Alexander Karabutov, Elena Cherepetskaya, Model-based measurement of internal geometry of solid parts with sub-PSF accuracy using laser-ultrasonic imaging. NDT and E International, 2019, Volume 105, pp. 56-63. doi: 10.1016 / j.ndteint.2019.05.006
  7. Compression test parameters added;
  8. The detector is called RTN-31, there was a blot with the name
  9. Stress versus strain is meant, right.
  10. Indeed, for sample no. 2 (not 1) it was destroyed with a deformation of ~ 0.7 mm, but the recording of the IR radiation intensity was stopped earlier due to the danger of pieces of the sample flying off during destruction. The assumed value of the fracture stress was obtained statistically for previously fractured specimens. The samples began to “crack” at deformation values ​​of 0.6 mm or more.
  11. Thank you for this comment. Indeed, a mistake was made when expressing formula 8 from formula 6. Now the formula is correct.
  12. For the first time, based on calculations of the P-wave velocity in a limestone sample, in its certain constituent parts of calcite and quartz in close-packed structures [Appendix A], by precision measurements of wave velocities, the values ​​of the closed and open density were calculated, and, accordingly, the total. And on the basis of these values, the heat capacities of the dry and wet sample were calculated and a relationship with temperature changes was found. The density values ​​of limestone samples were measured by the trivial method of the ratio of mass to volume of the sample, and the density of calcite was taken from the reference book. Elastic moduli were determined according to GOST 28985-91 (state all-Russian standard 28985-91), which is similar to ASTM 3148-02.

Reviewer 3 Report

For the Authors:

Really well written paper! Extensive review of bibliography, clear material and methods and sound discussion and conclusion. In my opinion, it only needs minor adjustments in the highlighted sections of the attached document. 

Wish you all the best, 

Your reviewer. 

Author Response

Thank you for your comments. The adjustments have been made. It is important to note that when editing an article, the numbering of the "lines" changed. Major fixes have been edited / added and are highlighted in green.

It is also important to note that the abstract was formulated with an emphasized wording of novelty and specific contributions from this article.

Minor adjustments were made at the suggestion of Reviewer # 3.

  1. Replaced terminology in some places and made technical adjustments.

Thank you for the errors and comments found, the article has become better.

Best regards, authors.